# MicroRNA (miRNA) Complexity in Alzheimer’s Disease (AD)

**DOI:** 10.3390/biology12060788

**Published:** 2023-05-30

**Authors:** Walter J. Lukiw

**Affiliations:** 1LSU Neuroscience Center, Louisiana State University Health Science Center, New Orleans, LA 70112, USA; wlukiw@lsuhsc.edu; 2Alchem Biotech Research, Toronto, ON M5S 1A8, Canada; 3Department of Ophthalmology, LSU Health Science Center, New Orleans, LA 70112, USA; 4Department Neurology, Louisiana State University Health Science Center, New Orleans, LA 70112, USA

**Keywords:** Alzheimer’s disease (AD), biomarker, circular RNA, exosomes (EX), extracellular microvesicles (EMVs), hippocampus, inflammatory neurodegeneration, lipopolysaccharide, miRNA-146a, RNA sequencing

## Abstract

**Simple Summary:**

Alzheimer’s disease (AD) is a progressive, age-related neurodegenerative disorder representing the most common cause of senile dementia and neurological dysfunction in our aging domestic population. MicroRNAs (miRNAs) are a small family of non-coding single-stranded RNAs (ssRNAs) that function in complex interactive networks to direct the post-transcriptional repression of messenger RNA (mRNA)-encoded genetic information. Current evidence suggests that these ssRNAs perform an important regulatory role in the expression of genes in the AD-affected brain and central nervous system (CNS). This review paper will focus on recent studies, developments and advancements in our appreciation and understanding of the complexity of miRNA signaling in AD-affected brain hippocampal CA1 compared to age- and gender-matched controls.

**Abstract:**

AD is a complex, progressive, age-related neurodegenerative disorder representing the most common cause of senile dementia and neurological dysfunction in our elderly domestic population. The widely observed heterogeneity of AD is a reflection of the complexity of the AD process itself and the altered molecular-genetic mechanisms operating in the diseased human brain and CNS. One of the key players in this complex regulation of gene expression in human pathological neurobiology are microRNAs (miRNAs) that, through their actions, shape the transcriptome of brain cells that normally associate with very high rates of genetic activity, gene transcription and messenger RNA (mRNA) generation. The analysis of miRNA populations and the characterization of their abundance, speciation and complexity can further provide valuable clues to our molecular-genetic understanding of the AD process, especially in the sporadic forms of this common brain disorder. Current in-depth analyses of high-quality AD and age- and gender-matched control brain tissues are providing pathophysiological miRNA-based signatures of AD that can serve as a basis for expanding our mechanistic understanding of this disorder and the future design of miRNA- and related RNA-based therapeutics. This focused review will consolidate the findings from multiple laboratories as to which are the most abundant miRNA species, both free and exosome-bound in the human brain and CNS, which miRNA species appear to be the most prominently affected by the AD process and review recent developments and advancements in our understanding of the complexity of miRNA signaling in the hippocampal CA1 region of AD-affected brains.

## 1. Introduction

MicroRNAs (miRNAs) represent a family of small, non-coding single-stranded RNAs (ssRNAs) that function in complex interactive networks to direct the post-transcriptional repression of messenger RNA (mRNA) encoded genetic information [1,2,3,4,5,6]. In the majority of cases, miRNAs interact via nucleotide complementarity within the 3′ untranslated region (3′-UTR) of their target mRNAs to induce degradation and translational repression of that mRNA [3,4,5,7,8]. Present in diverse eukaryotic lineages, miRNAs play critical regulatory roles in the control of gene expression and serve key functions in physiological, homeostatic and pathological processes, especially in the brain and CNS. miRNAs are particularly versatile epigenetic entities since: **(i)** a single miRNA can target multiple mRNAs, a single mRNA can be the target of multiple miRNAs and alterations in the abundance, speciation and complexity of a few critical miRNAs induce complex alterations in cellular homeostasis and phenotype; and **(ii)** miRNAs can therefore trigger the disruption of complex genetically based signaling networks associated with the occurrence of multiple age-related pathologies in *Homo sapiens* [3,4,7,8,9,10,11,12,13,14,15]. This review paper will focus on recent studies, developments and advancements in our understanding of the complexity of miRNA signaling in the hippocampal CA1 of AD-affected brain, an anatomical region of the human brain targeted by the AD process. 

## 2. miRNA Complexity in the Brain

As the most compact and smallest known carrier of highly specific genetic information, miRNAs have a number of fascinating molecular-genetic and information-carrying properties. Most miRNAs are ~22 nucleotide (nt) long ssRNAs with the possibility of four different ribonucleotides (A,G,C,U) at any of the 22 positions. This should generate ~4 to the power of 22 random miRNA species, which is 1.76 × 10^13^ different miRNAs. Using RNA sequencing, DNA and miRNA array-based analysis and advanced Northern quantitative techniques, the current accepted total maximum number of miRNAs in any human cell is probably only around ~2,650, so there has been a tremendous amount of selection pressure for only certain miRNA sequences to persist and fulfill specific physiological, neurological and molecular-genetic functions [16,17,18,19,20]. The actual number of abundant miRNAs in control brain tissues is about ~30–80 unique miRNA species [9,12,21,22,23,24]. Currently, only about ~31 distinct miRNA species appear to be dysregulated by the AD process and not all of their mean abundances are equally affected (**Table 1** and **Table 2**). Importantly, not all miRNAs are present at every stage of cellular development or stage of disease, nor are they present in each and every cell type, and cells, tissues, nanovesicles and circulating biofluids often show significant differences in miRNA abundance, speciation and stoichiometry [24,25,26,27]. 

**Table 1 biology-12-00788-t001:** Description of brains, brain tissue pools, age, gender and total RNA quality from hippocampal CA1 region (a neuroanatomical target for AD-type neuropathologcal change) utilized in these miRNA complexity studies.

Sample Designation	Age Range (yr)	Mean Age +/− 1 SD (yr)	Gender	PMI (hr)	18S:28S RNA Ratio	Total RNA Quality
**control group**	71–82	76.2+/−10.2	12 M, 24 F	1.6–3.5	2.0–2.2	2.0–2.1

**Legend to Table 1**- age range (yr) = range of tissue age (years) used in the current RNA study; +/− 1 SD = plus or minus one standard deviation; PMI = post-mortem interval (death to freezing of brain-tissue interval at −81 °C) in hr; numbers in the ‘18S:28S RNA ratio’ and ‘total RNA quality’ columns represent an index for the spectral purity of the RNA used; the ratio of spectral absorbance at 260 nm/280 nm > 2.0 indicates very-high-quality RNA; based on an *n* = 36; the use of RNase-free plastic-ware and extraction reagents containing RNAsecure (Ambion Inc., Austin, TX, USA) and/or RNaseOUT (ThermoFisher Scientific, Waltham, MA, USA) ribonuclease inhibitors yielded high spectral quality RNA as was analyzed using RNA LabChip Analysis Chips (Caliper Life Sciences Inc., Mountain View, CA, USA) and a 2100 Bioanalyzer (Agilent Technologies, Santa Clara, CA, USA); chips were analyzed using an LC Sciences miRNA array Genechip (proprietary MRA-1001-miRNA microfluidic chip analytical platform); in total 2,650 small RNAs were analyzed; LC Sciences Corporation, Houston, TX, USA; https://lcsciences.com/company/technology/technology-microarray/; (last accessed on 19 April 2023).

**Table 2 biology-12-00788-t002:** Relative abundance and speciation of miRNA in the *Homo sapiens* brain hippocampal CA1 region; miRNAs overlain in gray are altered in abundance in AD hippocampus; (see **Table 1**, **Figure 1** and manuscript text).

Let-7a **	miRNA-24	miRNA-99b	miRNA-151a	miRNA-361
Let-7b *	miRNA-26a	miRNA-100	miRNA-151b	miRNA-450b *^,f^
Let-7c **	miRNA-26b	miRNA-103	miRNA-155 *^,f^	miRNA-451a
Let-7d **	miRNA-27a	miRNA-107 *	miRNA-181a *	miRNA-485
Let-7e *	miRNA-27b *	miRNA-124a *	miRNA-181b	miRNA-638
Let-7f *	miRNA-28	miRNA-125a	miRNA-181d	miRNA-3613
Let-7g **	miRNA-29a	miRNA-125b **^,f^	miRNA-185	miRNA-3656
Let-7i **	miRNA-29b	miRNA-126	miRNA-191 *	miRNA-3665
miRNA-7 *^,f^	miRNA-29c	miRNA-127	miRNA-195	miRNA-3960
miRNA-9 *^,f^	miRNA-30b *^,f^	miRNA-128a *	miRNA-200	miRNA-4301
miRNA-15b	miRNA-30c	miRNA-128b *	miRNA-204	miRNA-4324
miRNA-16 *	miRNA-30d *	miRNA-132 *	miRNA-221	miRNA-4454
miRNA-21	miRNA-31	miRNA-143	miRNA-222 *	miRNA-4484
miRNA-23a	miRNA-34a *^,f^	miRNA-144	miRNA-223	miRNA-4516
miRNA-23b	miRNA-99a	miRNA-145 *	miRNA-320 *	miRNA-4787
miRNA-23c	miRNA-98	miRNA-146a *^,f^	miRNA-342 *	miRNA-5001

**Legend to Table 2**—This recently updated table indicates the most abundant unique miRNA species (*n* = 80) detected in the human brain hipppocampal CA1 region and includes miRNA expression levels complementary to data in previous reports [6,7,9,10,12,21,22,23,24,25,26,27,28,29,30,31]; importantly, miRNA species overlain in light gray (and their mRNA targets) have been strongly implicated in AD-relevant processes [9,32,33,34,35,36]; those miRNAs overlain with a darker shade of gray are further described in **Figure 1**; miRNAs with a single asterisk such as miRNA-7, miRNA-9, miRNA-30b, miRNA-34a, miRNA-125b, miRNA-146a, miRNA-155 and miRNA-450b are moderately abundant in controls; however, they increase several-fold in abundance in AD; many of these same miRNAs are implicated in other infectious, neuro-inflammatory, and/or immunological diseases including prion disease (PrD) [21,32,33,34,35,36,37,38,39,40,41,42]; miRNAs with two asterisks are among the most abundant miRNAs in this same region of the hippocampal CA1 region (see manuscript text); a superscript of **‘f’** denotes miRNAs known to be induced by the pro-inflammatory transcription factor NF-kB (p50/p65) complex [29,30,41,42]; abundances are categorized by miRNA numerical designation, based on pooled data from *n* = 36 short post-mortem interval (PMI) control tissues; data derived from [9,10,11,12,19,26,28,36,43,44,45,46,47,48,49,50]; the current group included 12 males and 24 females, mean age 76.2  ± 10.2 years; all post-mortem intervals (PMIs; death to brain freezing at −81°C) were 3.5 h or less (**Table 1**); each of these miRNAs yielded high (≥8000) units of signal strength on miRNA analytical array Genechips (proprietary MRA-1001-miRNA microfluidic chip analytical platform; in total the abundance of 2,650 miRNAs was analyzed; LC Sciences Corporation, Houston, TX, USA; https://lcsciences.com/company/technology/technology-microarray/ (last accessed on 19 April 2023); miRNAs were ranked as the 80 most abundant miRNAs detected; all other miRNAs are at a significantly lower abundance; note that there is a relatively high basal abundance of the let-7a to let-7i group of miRNAs that have been associated with AD as a group; we note that AD-abundant miRNAs such as the let-7 group, miRNA-7, miRNA-9, miRNA-21/23, miRNA-29a, miRNA-30b, miRNA-34a, miRNA-125b, miRNA-146a, miRNA-155 and miRNA-450b exhibit high relative variability in mean abundance in control brains and are significantly up-regulated in AD brains; other miRNAs may be involved; in human neural cells in primary culture, most of these same miRNAs are induced by reactive oxygen species (ROS) and/or the pro-inflammatory transcription factor NF-kB (p50/p65) [29,30,41,42,43]; interested readers are encouraged to access these recent documents for the latest information on individual miRNA complexity in AD and transgenic murine models of AD (TgAD); importantly, the miRNAs listed in **Table 2** should be useful in initiating studies of miRNA-mRNA linked signaling systems, networks and gene expression patterns during aging, AD and other forms of human brain neurodegeneration involving the hippocampal CA1 formation.

**Figure 1 biology-12-00788-f001:**
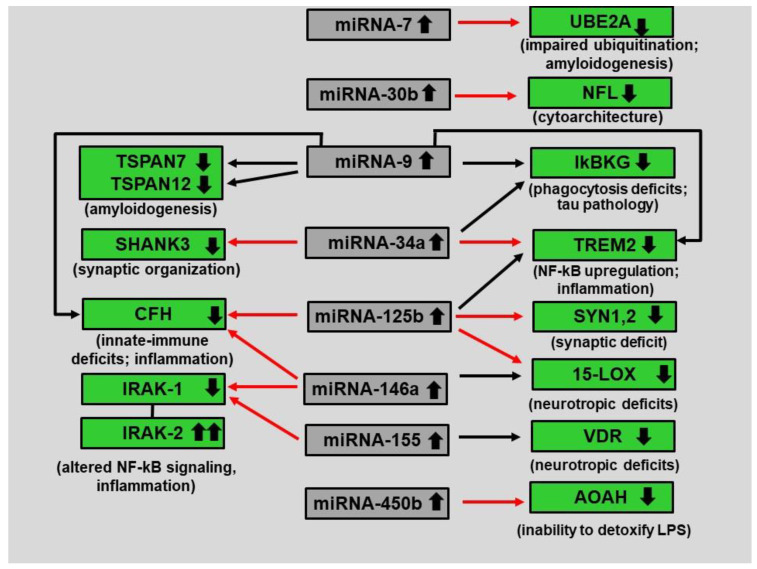
**Complex integration of miRNA–mRNA signaling in AD brain hippocampus. Legend to Figure 1:** This highly schematized figure indicates how the up-regulation of just eight miRNAs (central gray boxes; small black arrows pointing upward within the box) impacts the levels for 13 AD-relevant mRNAs (green boxes; small black arrows pointing downward within box) and can explain much of the neuropathology of AD; all eight miRNAs are under NF-kB regulatory control and all have been reported to be upregulated in AD brain tissues [29,30,31,43,44,45,46,47,48,49,50,51] (**Table 2**); all eight up-regulated miRNAs are highlighted in gray in **Table 2**; all long red arrows are established interactions; long black arrows represent predicted interaction(s); miRNAs predominantly act to decrease mRNA levels and different miRNAs may have multiple mRNA targets [8,10,11,12,15,25,34,51,52,53]; other miRNA-mRNA pairings may be involved; interestingly both miRNA-146a and miRNA-155 upregulation lead to a down-regulation of IRAK-1 expression with a compensatory surge in the expression of IRAK-2 that leads to altered NF-kB signaling and inflammation [48]; abbreviations: TSPAN7,12 = tetraspanin transmembrane proteins 7 and 12 [24]; SHANK3 = SH3 and multiple ankyrin repeat domain protein 3 [53]; CFH = complement factor H [7,12,17,35]; IRAK 1,2 = interleukin-1 receptor-associated kinase-1 [48]; UBE2A = ubiquitin conjugating enzyme 2A [54]; NFL = neurofilament light chain (involved in neuronal shape and cytoarchitecture) [55]; IkBKG = inhibitor of the nuclear factor kappa B kinase regulatory subunit gamma [29,30,55,56,57,58,59]; TREM2 = an amyloid sensing- and clearance receptor protein, the triggering receptor expressed in myeloid/microglial cells [56,57]; SYN1,2 = synapsin 1 and 2 (pre-synaptic proteins involved in neurotransmitter release) [12,17,58,59]; 15-LOX = the neurotrophic regulatory enzyme 15 lipoxygenase [60]; AOAH = a detoxifying enzyme responsible for inactivating Gram-negative bacterial-derived lipopolysaccharide (LPS) [61,62].

Interestingly: **(i)** many brain-and CNS-enriched miRNAs contain exactly the same 22 nt ribonucleotide sequence in mice, archaic primates and humans (divergence in evolution ranging from about ~10^6^ to 10^7^ years; see below), suggesting that these miRNAs, strongly conserved over long periods of time, are serving the same critical regulatory, genetic and physiological functions over vast periods of evolution [63,64]; **(ii)** eukaryotic miRNAs are remarkably similar in structure and function to plant viroids, autonomously self-replicating small ssRNAs that do not encode any protein, lack a capsid and are known to trigger disease in multiple plant species [18,19,20,65,66]; **(iii)** there has recently been described an intriguing possibility that miRNAs and/or ssRNAs derived from dietary sources have potential to contribute not only to intracellular but also inter-species signaling and, in doing so, modulate molecular-genetic mechanisms associated with human diet and dietary intake, health and the spread of transmissible disease [13,16,66,67,68]; and **(iv)** individual miRNA abundance and speciation varies widely among human individuals; the capability of miRNA as an ssRNA to target and inactivate lethal viruses such as the 29,811 nt SARS-CoV-2 ssRNA may help explain the wide variability in an individual’s intrinsic sensitivity and susceptibility to SARS-CoV-2 infection and COVID-19 disease and the development of neurological complications, especially during the post-COVID-19 recovery period [37,38,69,70,71]. 

## 3. miRNAs in Alzheimer’s Disease (AD)

Over the last ~16 years about ~2000 peer-reviewed research papers, including about ~600 review articles, have described the status of miRNA presence: **(i)** in AD-affected brain cells, tissues and biofluids compared to healthy age- and/or gender-matched controls; **(ii)** in human neuronal-glial (HNG) cells, mononuclear cells and other cell types in primary tissue culture stressed with cytokines (IL-1β, IL-6, TNFα), Aβ peptides, lipopolysaccharide (LPS), hypoxia and other AD-relevant stressors; and **(iii)** in transgenic murine models of AD (TgAD) such as the 5xFAD and other murine models into which specific AD-relevant genes have been inserted for the purpose of studying the mechanisms of their overexpression in the brain as these TgAD models age [7,8,9,10,11,12,13,14,15,16,17,18,19,28,29,30,31,32,33,34,35,36,39,40,41,42,43,44,45,46,47,48,49,50,52,72,73,74,75,76,77,78,79]. Many of these reports emphasize the finding of a single miRNA species significantly dysregulated in AD or in AD models and altered expression patterns in the transcriptome that are specifically targeted by a single abundance-modified miRNA. Overlapping familial, genetic, molecular, neuro-pathological and epidemiological studies and RNA-sequencing data continue to indicate that AD is an extremely heterogeneous and multifactorial neurological disorder. Current research findings further underscore the enormous genetic complexity of the human microRNA–mRNA interactome as reflected by the remarkable heterogeneity of the AD mechanism [17,23,44,75,77,78,80,81]. With all probability, the essential molecular-genetics, neurophysiology and neurobiology of every single case of AD is very different depending on patient age, age of AD onset, AD duration, gender, family history, AD diagnosis (and misdiagnosis), drug history, lifestyle, inter-current neurological and/or cardiovascular disease (for example overlap with multi-infarct dementia and cardiovascular and/or neurovascular disease), anatomical region of the brain and many other pathophysiological factors [43,44,45,46,47,49,50,51,52,53,54,55]. It is therefore highly unlikely that any one single miRNA can explain the totality of neurobiological dysfunction in an AD-affected brain; rather, a group of pathological miRNAs appears to be involved in the neuropathology of AD or other neurological syndromes [33,34,35,39,43,74,75,78,79,80,81,82,83] (**Table 1** and **Table 2**). While multiple therapeutic strategies involving anti-ssRNA and/or anti-miRNA (AM; antagomir) strategies have been proposed for the treatment of multiple human diseases ranging from Duchene muscular dystrophy (DMD) to non-Hodgkins lymphoma to brain and pancreatic cancers, no clinical trials are currently focusing on an AM approach for AD treatment, largely because of the significant intrinsic heterogeneity of this disease and its individual complexity [28,29,75,76]. It has recently become apparent that certain specific AM-directed therapeutic strategies may be potentially useful in individual AD cases using a ‘personalized’ or ‘precision medicine’ approach, custom-designed for each individual AD patient (see below) [74,78,80,81,82,83]. 

## 4. Specific Features of miRNA Abundance, Speciation and Trafficking in AD

The following selective points are applicable to our current understanding of miRNA abundance, speciation, complexity and trafficking in the AD-affected brain and CNS compared to age- and gender-matched controls:Multiple independent studies using Northern analysis with single or multiple radiolabeled or fluorescent probes, RNA sequencing and/or DNA, mRNA and/or miRNA array technologies indicate that gene expression as indexed by total messenger RNA (mRNA) abundance and yield is in general reduced in AD brains, probably a reflection of aging brain cells ‘shutting down’, and this correlates with a resultant inability to meet homeostatic demands and/or related molecular-genetic mechanisms in which total mRNA populations are found to be significantly reduced [45,47,84,85,86,87,88,89,90,91];As mammalian miRNAs predominantly act to decrease target mRNA levels, down-regulated gene expression indicates that down-regulated mRNA is, in part, probably the consequence of up-regulated miRNA as is widely observed in AD-affected brain tissues [34,35,48,49,52,73]; interestingly, many of the up-regulated miRNAs in AD are inducible and under transcriptional control of the pro-inflammatory transcription factor NF-kB (p50/p65) complex [29,30,35,52,73]; clearly, other inducible miRNAs and/or ROS and pro-inflammatory transcription factors may be involved;Importantly, no de novo appearance of any novel miRNAs has ever been observed in AD affected brains, only the up-regulation of existing miRNAs already present, and normally at homeostatic levels in age- and gender-matched control brain samples;Several independent studies have quantified and implicated the same brain-enriched miRNA species in the molecular-genetic processes involving innate-immune disruption, synaptic signaling deficits and inflammatory neurodegeneration as is observed in AD brains and verified in human brain cells, transgenic AD (TgAD) and other neurodegenerative disease models; these include at least 31 miRNAs all found to be significantly dysregulated in AD brains compared to age- and gender-matched controls from the same anatomical region; most appear to be up-regulated in mean relative abundance in anatomically-specific brain regions [5,6,7,8,9,10,11,12,13,14,15,17,18,19,20,21,22,23,28,29,31,32,33,34,35,36,37,38,39,40,41,43,44,46,48,49,50,67,68,69,70,71,72,73,74,75,77,78,79,80,83] (see **Table 1** and **Table 2**);One of the most-studied NF-kB (p50/p65)-induced and up-regulated miRNAs in AD brain neocortex and hippocampus is the 22 nt ssRNA pro-inflammatory and neuroimmune regulator miRNA-146a-5p (miRNA-146a) [30,35,36,37,39,41,48,49,77]; miRNA-146a, normally a brain- and CNS-abundant ssRNA, has been found to be significantly up-regulated in progressive and terminal viral- and prion disease (PrD)-mediated neurological disorders and in related neurological syndromes associated with inflammatory neurodegeneration, including at least eighteen different viral-induced encephalopathies for which data are available, in at least ten known PrDs of animals and humans, AD and in other sporadic and progressive age-related neurological diseases [36,37,39,40,41,49,77,92]. Despite the apparent absence of nucleic acids in prions, both DNA- and RNA-containing viruses (along with prions) significantly induce miRNA-146a in the infected host; however, whether this represents part of the host’s adaptive immunity, innate-immune response or a mechanism to enable the prion or virus a successful invasion of the host is still not well understood [30,37,38,39,41,91];As briefly mentioned earlier, another remarkable feature of the 22 nt AD-abundant miRNA-146a sequence (5′-ugagaacugaauuccauggguu-3′; encoded as a single copy gene at human chr 5q33.3) is its evolutionary conservation and persistence across *Homo sapiens, Rodentia* (rodents), *Muridae* (mouse), *Cervidae* (deer, elk), *Bovidae* (cattle), *Mustelidae* (mink) and other mammalian families in terms of anatomy, neurophysiology and molecular-genetic function [36,40,41,48,49,77]; this exact same miRNA-146a sequence has been conserved over an evolutionary divergence of about 9.6 × 10^7^ years between *Homo sapiens* and *Muridae* (*Mus musculus*; the common mouse) [63,64,92]; this suggests that miRNA-146a has been serving some well-conserved regulatory function in the post-transcriptional control of gene expression across many millions of years of evolution; it is further useful in the study of the molecular-genetic and epigenetic signaling that TgAD mice (and other AD murine models) and humans have a common miRNA-146a regulator that can be studied in both species as to their role in inflammatory neurodegeneration and the molecular-genetic mechanism of the AD process;Related to the point above, multiple research findings currently indicate a significant and highly interactive role for the NF-kB (p50/p65)-regulated miRNA-146a: **(i)** as a major small ssRNA regulator of innate-immune responses and inflammatory signaling in cells of the human brain and CNS; **(ii)** as a critical component of the complement system and immune-related neurological dysfunction with miRNA-146a involved in the down-regulation of complement factor H (CFH) [7,12,17,35]; **(iii)** as an inducible ssRNA in the brain and CNS that lies at a critical intersection of several important neurobiological adaptive immune response processes; **(iv)** as a potential biomarker for viral infection, transmissible spongiform encephalopathy (TSE) and AD and other neurological diseases in both animals and humans; **(v)** as a novel and unique ssRNA-based biomarker for inflammatory neurodegeneration in multiple species experiencing a decline in cognition, memory and other normal brain-specific functions; and **(vi)** as a regulator of the expression of the tetraspanin-12 (TSPAN-12; NET-2; TM4SF12), a four-pass integral transmembrane protein involved in both neurovascular development and amyloidogenesis [24,28,29,37,39,40];Several up-regulated miRNAs have a tendency to target the expression of AD-relevant mRNAs in AD brains and contribute to AD-relevant neurobiological processes; these specifically include synaptic and cytoskeletal deficits, the clearance of waste products from the cell, amyloidogenesis, tau pathology and neuro-inflammation, with accompanying insufficiencies and/or disturbances in innate-immunity, synaptogenesis, neuronal cytoarchitecture, phagocytosis, a progressive impairment in Aβ42 peptide clearance and disturbances in behavior, cognition and memory [12,32,79,93] (**Figure 1**);Very recent evidence suggests that miRNA translocation and trafficking in AD brains may in part be mediated by a network of lymph- and blood-borne nanovesicles (NV), exosomes (EXs) and/or extracellular microvesicles (EMVs) throughout highly vascularized brain tissues; NVs, EXs and EMVs are representative of a diverse collection of plasma membrane-derived nanovesicles, 30–1000 nm in diameter, released by all cell lineages of the human CNS; these nanovesicles are examples of a very active and dynamic form of extracellular communication and the conveyance of biological information transfer essential to maintain homeostatic neurological function; NVs, EXs and EMVs contain complex molecular cargoes that are representative of the cytoplasm of their cells of origin [94,95,96,97,98,99,100,101,102]; these include various mixtures of lipids, glycolipids, cytokines, chemokines, proteins, proteolipids, carbohydrates, polysaccharides, miRNAs, mRNAs, inflammatory mediators and other components, including end-stage neurotoxic and pathogenic metabolic products, such as tau fragments, lipopolysaccharide (LPS) and various amyloid beta (Aβ) peptides [91,95,96,97,98,99,100,101,102];Brain microglia respond to neurodegenerative diseases with complex reactions via the induction of a pro-inflammatory phenotype, and may secrete NVs, EXs and/or EMVs enriched in potentially pathogenic miRNAs such as the Let-7 series of miRNA-like ssRNAs, miRNA-7, miRNA-9, miRNA-30b, miRNA-34a, miRNA-125b, miRNA-146a, miRNA-155, miRNA-450b and others that are known to promote neuro-inflammation and amyloidogenesis, induce complement activation, disrupt innate-immune signaling and deregulate the expression of neuron-specific phosphoproteins involved in neurotropic support and synaptic signaling [98,99,100,101,102,103,104];As indicated earlier, while no de-novo-appearing miRNAs are currently associated with any neurological disease, complex patterns of a finite number dysregulated miRNAs have been associated with biofluids analyzed from AD at all disease stages; as such, this family of pathology-associated miRNA species in the brain and CNS may not be absolute definitive biomarkers for AD, but instead contribute to all-cause neurodegeneration, diagnosis, prognosis and therapeutic drug monitoring in a related family of neurodegenerative disorders [17,32,79,80,82];miRNA complexity data in AD may be most useful after its careful integration with other diagnostic modalities that include extended clinical observation, cognitive testing and neurological assessment, computerized axial tomography (CAT)-, magnetic resonance imaging (MRI)- and/or positron emission tomography (PET)-based brain imaging, bioinformatics and biostatistical analysis and predictive, preventive, precision and personalized (P4) medical approaches designed to optimize the disease trajectory for each individual AD patient [12,17,35,80,82];Human miRNAs, as a naturally occurring family of ssRNAs, can be physically linked together in tandem chains and often circularized into novel miRNA structures referred to as ‘circular RNAs’ (circRNAs) with miRNAs concatenated into a closed circular loop; first described just ~10 years ago [105], traditional methods of RNA detection, analysis and characterization requiring a free 3′ or 5′ ribonucleotide terminus may have significantly underestimated circRNA abundance and significance in eukaryotic cells [7,31,32,105,106]; intrinsically resistant to exonucleolytic RNA attack and decay, these circRNAs appear to be enriched in mammalian brain cells and CNS and retinal tissues [7,31,32,54,105,106]; circRNAs may occur either as single entities within brain cells or may be compartmentalized into NVs, EXs and/or EMVs (see above); specific ssRNAs such as the evolutionary ancient microRNA-7 (miRNA-7; chr 9q21.32; an important post-transcriptional regulator of human brain gene expression) are not only highly abundant in human brain cells but are also associated with a circRNA for miRNA-7 (ciRS-7) in the same tissues; ciRS-7 contains multiple, tandem anti-miRNA-7 sequences; ciRS-7 thereby acts as a kind of endogenous, competing, anti-complementary miRNA “sponge” to adsorb, and hence quench, normal miRNA-7 functions and activities [54,105,106]; deficits in ciRS-7, and ciRS-7 “sponging activities”, might be expected to increase ambient miRNA-7 levels in AD-affected brain cells, as is observed in AD-affected brains, to ultimately contribute to the down-regulation of selective miRNA-7-sensitive messenger RNA (mRNA) targets [54,105]; such miRNA-mRNA ‘sponging’ systems mediated by cell- and/or tissue-enriched circRNAs appear to represent another important layer of epigenetic control over gene expression at the post-transcriptional level in both health and disease [7,31,32,54,105,106];Recent evidence also indicates that resident microbes of the human gastrointestinal (GI)-tract microbiome have the potential to provide a life-long supply of microbial-derived neurotoxins including bacterial amyloids and RNA, glycolipids including lipopolysaccharide (LPS), and other potent endotoxins that appear to have significant and deleterious effects on brain miRNA complexity and which contribute to altered gene expression signaling in the AD brain [31,50,51,107,108,109,110];Lastly, it should be appreciated that while miRNAs appear to be playing critical roles in the post-transcriptional regulation of gene expression in health and disease and their activities highlight the enormous complexity of the human microRNA–mRNA interactome, currently, the majority of human miRNA–mRNA interactions remain unidentified and are under intensive study. The intrinsic complexity and magnitude of gene expression in the human brain and CNS make the elucidation of even fundamental miRNA-mRNA signaling pathways exceptionally challenging, as does the search for reliable biomarkers for neurodegeneration in the periphery. The manipulation of gene expression using miRNA-based strategies and the use of extracellular vesicles for therapeutic delivery should be of great strategic value in the clinical management of AD and related forms of progressive age-related inflammatory neurodegeneration [31,35,50,51,80,81,82,83,111,112,113,114,115,116,117,118,119,120].

## 5. Targeting miRNA for AD Treatment and Related Therapeutic Strategies

Neurological disorders of *Homo sapiens* are generally complex, progressive, and insidious, and AD represents the most common form of cognitive and memory impairment in our aging domestic population. It has been about ~117 years since the first description of AD, and currently no adequate or effective treatment exists for this ultimately lethal neurodegeneration of the human brain and CNS [6]. The idea that small non-coding RNAs (sncRNAs) and/or miRNAs were centrally involved in the etiopathogenesis of AD and progressive, age-related human neurodegenerative disorders was first proposed about 25 years ago, however it was not until 2007 that specific miRNA abundance, speciation and localization to the hippocampal CA1 region (an anatomical area of the human CNS specifically targeted by the AD process) was shown to strongly associate with AD-type change when compared to age- and gender-matched neurologically normal controls [9,72]. The now well documented obvious importance of a global disruption in miRNA signaling in the AD-affected brain currently suggests that blocking or modulating miRNA abundance or the transcription factors such as the NF-kB p50/p65 complex that regulates miRNA generation may have some therapeutic value [5,7,9,12,26,30,37]. To this end a number of stabilized anti-miRNA (AM; antogomir), sense-RNA and anti-sense RNA-based oligonucleotide strategies have been proposed as novel therapeutic agents in the clinical management of AD [28,29,76,79,107,108,109,110,111,112,113,114]. The major problem using these types of approaches are: **(i)** the extreme heterogeneity of AD onset, neuropathology and disease course among individual AD patients; and **(ii)** the fact that miRNAs and the transcription factors that regulate them (such as the pro-inflammatory dimeric NF-kB p50/p65 complex) have both multiple and complicated off-target effects [28,37,39,112,113,114]. No miRNA-based therapeutic approaches have yet been successful in the clinical management of AD. However, several recent proposed therapeutic approaches may be in the implementation of the bundling of miRNAs, AM and other AD-relevant relevant therapeutic components, including inflammatory mediators into nanovesicle (NV), exosome (EX) and/or microvesicle packets tailored to individual AD patients and guided by the use of precision and/or personalized medical approaches [28,98,100,102,114].

## 6. Discussion

AD is a slow, insidious and progressive neurological dysfunction characterized by the loss of neurons and synapses in the brain and CNS that is now affecting hundreds of millions of elderly people worldwide. This irreversible, age-related terminal disorder has been one of the most extensively studied neurological disorders and one of the most perplexing and difficult to diagnose in the history of contemporary neurology. A major complication in the diagnosis of AD is its intrinsic complexity, disease heterogeneity and inter-current age-related neurological disorders that accompany AD, including common age-related dysfunction of the neural vasculature. The biochemical and neurobiological nature of miRNAs indicate that their study can provide very valuable neurological and molecular-genetic information in the elucidation of functional miRNA–mRNA-linked signaling networks in both healthy aging and in age-related neurological disease. miRNA analysis and characterization can further provide valuable clues to our mechanistic understanding of the AD process, especially in the sporadic forms of this complex brain disorder. Current in-depth analyses of high-quality AD and age- and gender-matched control brain tissues and those using transgenic AD (TgAD) models are providing pathophysiological and epigenetic miRNA-based signatures of AD that can serve as a basis for the future design of effective RNA-based treatments. RNA-based medications can include stabilized miRNAs, anti-miRNA (AM, antagomir) and other approaches directed against specific miRNA activators such as NF-kB (p50/p65), using targeted anti-NF-kB (p50/p65) treatment strategies, or any combination of these advanced therapeutics. NV, EX and/or EMV packets containing strategically designed and stabilized miRNA, anti-miRNA (AM), mRNA in conjunction with other therapeutic components and inflammatory mediators and tailored to individual AD patients using precision medicine and predictive, participatory, preventive and personalized (P4) approaches should be useful in the clinical management of AD and other complex neurological disorders that are now urgently requiring effective disease-modifying intervention [17,35,53,55,56,57,58,59,60,61,62,75,79,80,81,82,100,112,113,114,115,116,117,118,119,120].

## Data Availability

All data used in this review are openly available and freely accessible on MedLine (www.ncbi.nlm.nih.gov; last accessed on 19 April 2023) where they are listed by the last names of the individual authors.

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
