# Peer review of "MicroRNA (miRNA) Complexity in Alzheimer’s Disease (AD)"

_biology, 2023, doi:10.3390/biology12060788_

Round 1

Reviewer 1 Report

miRNA complexity in AD

This review covers developments in the complexity of miRNA signaling in Alzheimer’s disease.  It is done so in a fairly concise and clearly written manner.  This topic is complex as evidenced by the 600 review articles already turned to this topic.

There are two main points.

First, some explanation of the search approach used should be included to provide detail of the literature covered.  The author claims that the review is “focused” and on “recent” developments.

Second, The Tables have lengthy legends, but these are not clear on major points.  The data seemingly derives from control tissue for both Tables and yet, in Table 2 and the text, there is discussion about miRNAs upregulated in AD.  The review purports to describe the role of miRNAs in AD and so this is a major part of the presentation. The highlight of miRNAs in grey as being upregulated in AD needs to be stated in the title and made more obvious in the legend.  The title also states “relative abundance”, when the table really shows the most abundant species in (control) brain tissue.

There were a few trivial typos noted:

178:  feture

185:  Mus musculus  (l.c.)

Table 2: 392:  sapiens

Table 2:  In addition to the points noted above, the meaning of *, ** and f should be made more clear rather than being buried within the very long legend.

Ref 5:  Barbato, C. /(MicroRNA

Ref 12:  Lukiw. W/J.. 

Ref 98. 95. Yin

Author Response

Please see appended file:

Reviewer 2 Report

In the manuscript entitled “microRNA (miRNA) complexity in Alzheimer’s disease (AD)”, the authors reviewed the role of miRNA in AD. This manuscript, in general, is interesting and well-written; however, I believe that it should be thoroughly modified and updated because of the following:

1.     The authors are encouraged to propose at least two figures to explain their message better; one should focus on the role of miRNA within the progression of AD in the brain and CNS.

2.     Another schematic representation of an AD-relevant (miRNA-mRNA) regulatory network involving various miRNAs and the molecular-genetic mechanism of microRNA (miRNA) generation in CNS. Please see the articles for reference-

doi:  10.3389/fnins.2020.585432

doi: 10.3389/fnmol.2020.00160

3.     The authors should also add a section of miRNAs' clinical aspects that help detect AD.

4.     Authors should also discuss the microRNAs as therapeutic Strategies in another section to highlight the need for the study or unmet medical needs.

5.     There are many grammatical errors and repetition of Alzheimer’s disease (AD) in the manuscript. Authors are requested to remove all such errors. 

Moderate editing required.

Author Response

Please see appended file

Round 2

Reviewer 2 Report

The authors have addressed all my concerns; the Manuscript is suitable for publication in its current form.